# Optimal Microphone Array Placement Design Using the Bayesian Optimization Method

**DOI:** 10.3390/s24082434

**Published:** 2024-04-10

**Authors:** Yuhan Zhang, Zhibao Li, Ka Fai Cedric Yiu

**Affiliations:** 1Department of Applied Mathematics, The Hong Kong Polytechnic University, Hung Hom, Kowloon, Hong Kong, China; yhmath.zhang@gmail.com; 2School of Mathematics and Statistics, Central South University, Changsha 410083, China; zblimath@csu.edu.cn

**Keywords:** Bayesian optimization, beamformer design, microphone placement, Gaussian process regression, acquisition function

## Abstract

In addition to the filter coefficients, the location of the microphone array is a crucial factor in improving the overall performance of a beamformer. The optimal microphone array placement can considerably enhance speech quality. However, the optimization problem with microphone configuration variables is non-convex and highly non-linear. Heuristic algorithms that are frequently employed take a long time and have a chance of missing the optimal microphone array placement design. We extend the Bayesian optimization method to solve the microphone array configuration design problem. The proposed Bayesian optimization method does not depend on gradient and Hessian approximations and makes use of all the information available from prior evaluations. Furthermore, Gaussian process regression and acquisition functions make up the Bayesian optimization method. The objective function is given a prior probabilistic model through Gaussian process regression, which exploits this model while integrating out uncertainty. The acquisition function is adopted to decide the next placement point based upon the incumbent optimum with the posterior distribution. Numerical experiments have demonstrated that the Bayesian optimization method could find a similar or better microphone array placement compared with the hybrid descent method and computational time is significantly reduced. Our proposed method is at least four times faster than the hybrid descent method to find the optimal microphone array configuration from the numerical results.

## 1. Introduction

Beamforming techniques can effectively obtain the sound of interest via spatial filtering to reduce interference and ambient noise from a mixed signal received by a set of microphone arrays. They are now widely used in the fields of wireless communications, hearing aids, and speech recognition [1,2,3,4]. Many techniques currently exist for solving the filter coefficients to achieve speech enhancement under specific conditions; for example, the linearly constrained minimum variance (LCMV) beamformer presented in [5] minimizes the power of the background noise, and dereverberation and interference suppressing are employed as constraints. It is worth noting that the length of the filters and number of microphones also greatly affect the beamformer’s performance. When filters reach a certain length, the performance limit enters stagnation and is far from satisfactory; in contrast, as the number of microphones is increased, the desired directivity pattern can be achieved under some circumstances [6]. In addition, the design of the microphone array’s location has a big influence on how well the beamformer works. Regular microphone array placements are always chosen [7], but it was discovered that the optimized microphone array placement within specific dimensions and areas significantly increased the overall performance compared to the regular placement in [8,9].

Many optimization problems and algorithms have been established to solve the microphone array configuration issue. The array-thinning technique [10,11] carefully selects the location of the microphone array while using fewer microphones to preserve the prior performance in the one-dimensional situation. Several studies have employed heuristic methods to identify the global optimal solution since the problem of optimization is non-convex and nonlinear. These include evolutionary programming [12], genetic algorithm (GA) [13,14,15,16], simulated annealing algorithm [17], pattern-search algorithm [18], and differential evolution [19]. However, these methods are often very time-consuming and have the risk of missing the optimal solution. In [20], an approach based on compressive sensing is described for building wideband sparse microphone arrays. The Taguchi method makes an effort to perform systematic experiments based upon orthogonal arrays to analyze the microphone design after pre-selecting multiple possible positions for the microphone elements [21]. By making the filter length sufficiently long, a nonlinear optimization problem on filter coefficients and microphone array placement using the $l_{2}$ norm is presented in [8]. This problem is then reduced to one where the placement variable is the only decision variable in [9], and a hybrid descent method incorporating a genetic algorithm is provided to obtain a more general solution.

Many methods mentioned above based on heuristic algorithms tend to be very inefficient. However, a Bayesian optimization method makes good use of the prior information from previous iterations and provides a posterior probability distribution to describe potential microphone array locations. The computational efficiency can be greatly improved. Bayesian optimization [22,23] is mainly for independent variables over continuous domains. It is widely applied in machine learning [24], the design of mechanical systems and materials [25,26,27], and the development of pharmaceuticals [28,29] because it is capable of tackling optimization problems with complicated objective functions.

This paper aims to extend the Bayesian optimization method to solve the microphone array configuration problem to improve the computational efficiency. Given that the non-convexity and non-linearity of the microphone array configuration optimization issue and the objective function form a dual integral, using heuristic methods is inefficient. The Bayesian optimization method can make full use of previously evaluated information by employing Gaussian progress (GP) regression to proxy the objective function of the microphone array design problem, while the acquisition function directs iterations to the point with the highest probability of being the minimum value.

In this study, we applied the Bayesian optimization method to find the optimal microphone array placement since the optimization problem with respect to the microphone array variables is non-convex and takes a long time to compute. GP regression can be applied to approximate the objective function and obtain the posterior distributions for the rest of the points in the feasible domain. The next sampling point should be smaller than the current minimum with greater probability and improvement by optimizing the acquisition function (obtained from the posterior probability distribution). As more configuration samples are evaluated, the posterior distribution is continually updated. This can bring the new feasible solution closer to the optimal locations for the microphone array. This method can efficiently achieve more excellent performance than the hybrid descent method with higher computational efficiency.

Our contributions can be summarized as the following:Considering that the microphone array design problem is non-convex and non-linear, the Bayesian optimization method is extended to solve the microphone array placement design problem;GP regression is used to surrogate objective function in the microphone array placement design optimization problem, while different acquisition function strategies are applied;Numerical experiments demonstrate that the proposed Bayesian optimization method could produce the same or better performance with shorter computational time compared with the hybrid descent method [9].

## 2. Problem Formulation

Assume that the signal received by each element of an array with *N* microphones is processed by a finite impulse response (FIR) filter and that *L* is the length of filter. The transfer function from the sound source to the *i*th microphone can be given as
$$A_{i}\left(r_{i},s,f\right)=\frac{1}{∥s-r_{i}∥}e^{\frac{-j2\pi f∥s-r_{i}∥}{c}},$$
the microphone has been fixed in $r_{i},i=1,\ldots ,M$, the location of the sound signal is determined by s, its frequency is determined by *f*, and *c* denotes the speed of sound in air. These FIR filter frequency responses are as follows when the signals are sampled synchronously at a rate of $f_{s}$ per second:$$W_{i}\left(w_{i},f,L\right)=w_{i}^{T}d_{0}\left(f,L\right),$$
where $w_{i}={\left[w_{i}\left(0\right),w_{i}\left(1\right),\ldots ,w_{i}\left(L-1\right)\right]}^{T}$ indicates the coefficients of the *i*th FIR filter, and the vector $d_{0}\left(f,L\right)$ is stated as
$$d_{0}\left(f,L\right)={\left[1,e^{\frac{-j2\pi f}{f_{s}}},\ldots ,e^{\frac{-j2\pi f}{f_{s}}\left(L-1\right)}\right]}^{T},$$
in which ${\left(\cdot \right)}^{T}$ denotes the matrix transpose.

Figure 1 illustrates the structure of the microphone array. The actual response by the beamformer could be constructed as follows based on the *i*th frequency response and the transfer function to the *i*th element microphone:(1)$$G\left(\lambda ,s,w,L\right)=\sum_{i=1}^{M}A_{i}\left(r_{i},s,f\right)W_{i}\left(w_{i},f,L\right),$$
where $\lambda =\{r_{1},r_{2},\cdots ,r_{M}\}$ is the set of microphone array locations and $w=\{w_{1},w_{2},\cdots ,w_{M}\}$ denotes the coefficients of all FIR filters.

As has been proven in Lemma 1 in [8], infinite-length filters and frequency-response functions have an equal relationship according to the infinite-length technique. In addition, the infinite-length technique has been applied in [8,9] to provide a beamforming output that is independent of the filter length *L*:$$\tilde{G}\left(\lambda ,s,\tilde{w},f\right)=\sum_{i=1}^{M}A_{i}\left(r_{i},s,f\right){\tilde{W}}_{i}\left({\tilde{w}}_{i},f\right),$$
where ${\tilde{w}}_{i}\in \tilde{\Gamma },\tilde{\Gamma }=\{\tilde{u}\left(f\right)+j\tilde{v}\left(f\right):\tilde{u}\left(f\right)$ and $\tilde{v}\left(f\right)$ are continuous and absolutely integrable, and the right-hand and left-hand derivatives exist, $\tilde{v}\left(0\right)=0$, $\tilde{v}\left(fs/2\right)=0$}.

Given that the desired response is $G_{d}\left(\lambda ,s,f\right)$ and that the $l_{2}$ norm is frequently employed as a measure of the error between $\tilde{G}\left(\lambda ,s,\tilde{w},f\right)$ and $G_{d}\left(\lambda ,s,f\right)$, the objective function with regard to the coefficients of beamformer $\tilde{w}$ and the microphone configuration variables λ are
$$F\left(\lambda ,\tilde{w}\right)=\frac{1}{\left|\Omega \right|}\int_{\Omega }\rho \left(\lambda ,f\right){\left|\tilde{G}\left(\lambda ,s,\tilde{w},f\right)-G_{d}\left(\lambda ,s,f\right)\right|}^{2}dsdf,$$
where Ω is a predefined spatial–frequency domain. ρ(λ,f) is a positive weighting function. The domain Ω is frequently composed of passband area $\Omega_{p}$ and stopband area $\Omega_{s}$. The following optimization problem determines a set of microphone array placements λ and a set of beamformer coefficients $\tilde{w}$ that minimize the error:(2)$$\begin{array}{c} \min_{\tilde{w}\in {\tilde{\Gamma }}^{N},\lambda \in \Lambda }\;F\left(\lambda ,\tilde{w}\right)\\ s.t.\;∥r_{i}-r_{j}{∥}^{2}\;\geq \;{\bar{\varepsilon }}_{d},i\neq j, \end{array}$$
Λ indicates the possible area for the microphone array. ${\bar{\varepsilon }}_{d}$ is the square of the minimum distance between two independent microphone elements, and restrictions $∥r_{i}-r_{j}{∥}^{2}\;\geq \;{\bar{\varepsilon }}_{d},i\neq j$ ensure that microphone elements work efficiently at a minimum distance from each other.

It is challenging to solve the non-convex optimization problem in Equation (Equation 2) as a whole since it consists of two separate kinds of variables. If the microphone array configuration is determined, the beamformer coefficient design is reduced to a convex optimization problem. Therefore, the optimization problem in (Equation 2) might be rewritten as
(3)$$\begin{array}{cc} \; & \min_{\lambda \in \Lambda }\;F\left(\lambda ,{\tilde{w}}^{*}\right)\\ \; & s.t.\;∥r_{i}-r_{j}{∥}^{2}\geq {\bar{\varepsilon }}_{d},i\neq j. \end{array}$$
The optimum beamformer coefficients under the specified array placements are ${\tilde{w}}^{*}$. However, the only decision variables λ nested inside $A_{i}\left(r_{i},s,f\right)$, $G_{d}\left(\lambda ,s,f\right)$, and the objective function in (Equation 3) are non-convex with regard to λ. Although the hybrid descent method proposed in [9] can find the optimal set of microphone array placements, its evaluation of the next microphone array location λ requires considerable time and results in an inefficient algorithm; a Bayesian optimization method is introduced to improve computational efficiency and to find a better microphone array configuration.

## 3. Bayesian Optimization Method

The non-convex optimization problem in (Equation 3) for the location variables λ is an extremely difficult to compute integral objective function. But, in the Bayesian optimization method, the multiple integral function in (Equation 2) might be demonstrated with a GP model. Moreover, the Bayesian optimization method, as has been proven by [23], can finally converge to the global optimal solution. A detailed description of the Bayesian optimization method for solving microphone array configuration problems is presented.

### 3.1. GP Regression

GP [30,31] has been considered as a good way to model loss functions in Bayesian statistical methods and has been applied in classification [32], face recognition [33], and neural networks [34]. Suppose that a finite collection of *n* different placements $\lambda_{1:n}$ is selected and that the objective function values and noisy observations are denoted by the variables $F(\lambda_{1:n})$ and $\bar{F}\left(\lambda_{1:n}\right)$, respectively. In GP regression, it is assumed that $F(\lambda_{1:n})$ will follow the a priori GP distribution and observation error $\varepsilon ∼N(0,\sigma^{2})$, resulting in observations $\bar{F}\left(\lambda \right)=F\left(\lambda \right)+\varepsilon$. Let $D_{n}={\left\{\left(\lambda_{i},\bar{F}\left(\lambda_{i}\right)\right)\right\}}_{i=1}^{n}$ denote the group of observations
(4)$$\bar{F}\left(\lambda_{1:n}\right)∼N\left(m_{0}\left(\lambda_{1:n}\right),\Sigma_{0}\left(\lambda_{1:n},\lambda_{1:n}\right)\right),$$
where
(5)$$\begin{array}{cc} \lambda_{1:n}= & [\lambda_{1},\lambda_{2},\ldots ,\lambda_{n}],\\ \bar{F}\left(\lambda_{1:n}\right)= & [\bar{F}\left(\lambda_{1}\right),\bar{F}\left(\lambda_{2}\right),\ldots ,\bar{F}\left(\lambda_{n}\right)],\\ m_{0}\left(\lambda_{1:n}\right)= & [m_{0}\left(\lambda_{1}\right),m_{0}\left(\lambda_{2}\right),\ldots ,m_{0}\left(\lambda_{n}\right)],\\ \Sigma_{0}\left(\lambda_{1:n},\lambda_{1:n}\right)= & \left[\begin{array}{ccc} Q(\lambda_{1},\lambda_{1}) & \cdots & Q(\lambda_{1},\lambda_{n})\\ \vdots & \ddots & \cdots \\ Q(\lambda_{n},\lambda_{1}) & \cdots & Q(\lambda_{n},\lambda_{n}) \end{array}\right]+\sigma^{2}I, \end{array}$$
Equation (Equation 5) is from [22]. $N(m_{0}\left(\cdot \right),\Sigma_{0}\left(\cdot \right))$ denotes a Gaussian prior distribution with $m_{0}\left(\cdot \right):R^{3\times 1}\mapsto R$ as the prior mean and $\Sigma_{0}\left(\cdot \right)\in R^{n\times n}$ as the covariance matrix. $Q(\lambda ,\hat{\lambda })$ is a kernel to measure the correlation of λ and $\hat{\lambda }$. Generally, the squared exponential kernel
(6)$$Q\left(\lambda ,\hat{\lambda }\right)=\sigma_{f}^{2}e^{\left(-\frac{∥\lambda -\hat{\lambda }{∥}^{2}}{2l^{2}}\right)}$$
is used. Equation (Equation 6) is from [32]. Notice that, the closer the two points, the bigger the value of the function, and that, the further away, the smaller the value of the function. This property shows that squared exponential functions are suitable to characterize the similarity between different microphone configurations.

### 3.2. Choosing Prior Hyperparameters

In the covariance matrix $\Sigma_{0}\left(\lambda_{1:n},\lambda_{1:n}\right)$, prior hyperparameters $\theta =:\{\sigma_{f},l,\sigma \}$ must be chosen in accordance with the provided observations samples $D_{n}$. Maximum likelihood estimation (MLE) is frequently used in probability and statistics to fit the GP model [35]. The distribution under these previous hyperparameters is known to us:$$\bar{F}\left(\lambda_{1:n}\right)|\theta ∼N\left(m_{0}\left(\lambda_{1:n}\right),\Sigma_{0}\left(\lambda_{1:n},\lambda_{1:n}\right)\right),$$
where we modify the notation in (Equation 4) to show that it depends on θ. The log-likelihood function can be easily obtained:(7)$$\log P\left(\bar{F}\left(\lambda_{1:n}\right)|\theta \right)=-\frac{1}{2}\log\left|\Sigma_{0}\right|-\frac{1}{2}{\bar{F}}^{T}\left(\lambda_{1:n}\right)\Sigma_{0}\bar{F}\left(\lambda_{1:n}\right)-\frac{n}{2}\log\left(2\pi \right).$$
Equation (Equation 7) is from [23]. Subsequently, the maximum likelihood function is employed to determine the prior hyperparameters:(8)$$\hat{\theta }=\arg\max_{\theta }\log P\left(\bar{F}\left(\lambda_{1:n}\right)|\theta \right).$$

### 3.3. Acquisition Function

Bayes’ rules predict that the random variable $\bar{F}\left(\bar{\lambda }\right)$ is normally distributed. The posterior mean and variance function are as follows: (9)$$\bar{F}\left(\bar{\lambda }\right)|\bar{F}\left(\lambda_{1:n}\right)∼N\left(m\left(\bar{\lambda }\right),\sigma^{2}\left(\bar{\lambda }\right)\right),$$
where
(10)$$\begin{array}{cc} m(\bar{\lambda }) & =\Sigma_{0}\left(\bar{\lambda },\lambda_{1:n}\right)\Sigma_{0}{\left(\lambda_{1:n},\lambda_{1:n}\right)}^{-1}\left(\bar{F}\left(\lambda_{1:n}\right)-m_{0}\left(\lambda_{1:n}\right)\right)+m_{0}\left(\bar{\lambda }\right),\\ \sigma_{n}^{2}\left(\bar{\lambda }\right) & =\Sigma_{0}\left(\bar{\lambda },\bar{\lambda }\right)-\Sigma_{0}\left(\bar{\lambda },\lambda_{1:n}\right)\Sigma_{0}{\left(\lambda_{1:n},\lambda_{1:n}\right)}^{-1}\Sigma_{0}\left(\lambda_{1:n},\bar{\lambda }\right), \end{array}$$
Equation (Equation 10) is from [22]. The input data $\bar{F}\left(\lambda_{1:n}\right)$ and prior $m_{0}\left(\bar{\lambda }\right)$ are averaged jointly to obtain the posterior mean $m(\bar{\lambda })$, whose weight is dependent on the kernel. The data provide more information; it should be shown that the posterior variance is always less than the previous variance.

The objective function’s prediction and uncertainty are represented by the posterior mean $m(\bar{\lambda })$ and variance $\sigma_{n}^{2}\left(\bar{\lambda }\right)$, calculated at each point $\bar{\lambda }$ in (Equation 10). The acquisition function has the responsibility of directing the pursuit of the optimum by these posterior functions. To locate the new sample placement, conventional improvement-based and optimistic acquisition methods are introduced.

The earliest acquisition function makes the next placement candidate superior to the optimal incumbent ${\bar{F}}_{n}^{*}=\min_{m\leq n}\bar{F}\left(\lambda_{1:m}\right)$ by maximizing the probability of improvement (PI) [36]:(11)$$\lambda_{n+1}=\arg\max_{\bar{\lambda }}PI\left(\bar{\lambda }\right),$$
where
$$\begin{array}{cc} PI(\bar{\lambda }): & =P(\bar{F}\left(\bar{\lambda }\right)<{\bar{F}}_{n}^{*})\\ \; & =\Phi (\frac{m\left(\bar{\lambda }\right)-{\bar{F}}_{n}^{*}}{\sigma_{n}\left(\bar{\lambda }\right)}). \end{array}$$
The standard normal cumulative distribution function is denoted by Φ(·). The posterior distribution of $\bar{F}\left(\bar{\lambda }\right)$ is expressed as in (Equation 9). The new point $\lambda_{n+1}$ is intended to have a high probability of being larger than the optimal incumbent ${\bar{F}}_{n}^{*}$, which will miss the point with larger gain but lower certainty.

Expected improvement (EI) [26,37] considers both the probability of improvement and the quantity of improvement. Suppose that we compute one of the remaining points $\lambda_{n+1}$ and the corresponding values $\bar{F}\left(\lambda_{n+1}\right)$ in the subsequent iterations; the optimal function value is either ${\bar{F}}_{n}^{*}$ or $\bar{F}\left(\lambda_{n+1}\right)$. If this quantity $\left[{\bar{F}}_{n}^{*}-\bar{F}\left(\lambda_{n+1}\right)\right]$ is positive, the improvement in the best observed point is $\left[{\bar{F}}_{n}^{*}-\bar{F}\left(\lambda_{n+1}\right)\right]$; if not, it is 0. This improvement could be expressed more succinctly as ${\left[{\bar{F}}_{n}^{*}-\bar{F}\left(\lambda_{n+1}\right)\right]}^{+}$, where $a^{+}:=\max\left(a;0\right)$ represents the positive part.

Since $\bar{F}\left(\lambda_{n+1}\right)$ is unknown, we can maximize the expected value of the improvement to make both this improvement ${\left[{\bar{F}}_{n}^{*}-\bar{F}\left(\lambda_{n+1}\right)\right]}^{+}$ and the possibility $P\left(\bar{F}\left(\lambda_{n+1}\right)<{\bar{F}}_{n}^{*}\right)$ large in the next point $\lambda_{n+1}$:(12)$$\lambda_{n+1}=\arg\max_{\bar{\lambda }}EI_{n}\left(\bar{\lambda }\right),$$
where
(13)$$\begin{array}{cc} EI_{n}\left(\bar{\lambda }\right):= & E_{n}\left[{\left[{\bar{F}}_{n}^{*}-\bar{F}\left(\bar{\lambda }\right)\right]}^{+}\right]\\ = & \left\{\begin{array}{cc} \sigma_{n}\left(\bar{\lambda }\right)\phi \left(\frac{\Delta_{n}\left(\bar{\lambda }\right)}{\sigma_{n}\left(\bar{\lambda }\right)}\right)+\Delta_{n}\left(\bar{\lambda }\right)\Phi \left(\frac{\Delta_{n}\left(\bar{\lambda }\right)}{\sigma_{n}\left(\bar{\lambda }\right)}\right), & \mathrm{if}\;\sigma_{n}\left(\bar{r}\right)>0,\\ 0, & \mathrm{if}\;\sigma_{n}\left(\bar{\lambda }\right)=0, \end{array}\right. \end{array}$$
denotes the expectation provided under the posterior distribution given observations $D_{n}$. Equation (Equation 13) is from [38]. If $\Delta_{n}\left(\bar{\lambda }\right)=m\left(\bar{\lambda }\right)-{\bar{F}}_{n}^{*}$ is the expected difference between the mean of the new point $\bar{r}$ and the previous best ${\bar{F}}_{n}^{*}$, then $E_{n}\left[{\left[{\bar{F}}_{n}^{*}-\bar{F}\left(\bar{r}\right)\right]}^{+}\right]$ is the expected value of improvement. The standard normal probability density function is denoted by ϕ(·).

The lower confidence bound (LCB) [39] strategy is widely applied in the field of multi-armed bandit [40]. Since the remaining points obey a Gaussian distribution $N(m\left(\bar{\lambda }\right),\sigma^{2}\left(\bar{\lambda }\right))$ and we want to find the minimum value of the objective functions in (Equation 3), the confidence lower bound can be expressed as
$$LCB\left(\bar{\lambda }\right):=m\left(\bar{\lambda }\right)-\beta \sigma \left(\bar{\lambda }\right).$$
To balance the mean and variance, the hyperparameter β is employed. Choose the next sampling point by minimizing $LCB(\bar{\lambda })$:(14)$$\lambda_{n+1}=\arg\min_{\bar{\lambda }}LCB(\bar{\lambda }).$$

A formal statement of Bayesian optimization to solve the microphone array placement location problem based on the design broadband beamformer is given in Algorithm 1.
**Algorithm 1** Bayesian optimization method for microphone array placement design**Initial step.** Select sensor location sample $r_{1},r_{2},\ldots ,r_{n}$, calculate objective function $F\left(r_{1}\right),F\left(r_{1}\right),\ldots ,F\left(r_{n}\right)$, prior mean value m(λ)=0 and set t=n.**S1.** Choose prior hyperparameters $\theta =:\{\sigma_{f},l,\sigma \}$ by MLE in (Equation 8) and $\bar{F}\left(r_{1}\right),\bar{F}\left(r_{2}\right),\ldots ,\bar{F}\left(r_{t}\right)$
can be obtained.**S2.** The resulting prior distribution on $\bar{F}\left(\lambda_{1}\right),\bar{F}\left(\lambda_{2}\right),\ldots ,\bar{F}\left(\lambda_{t}\right)$ is
$$\bar{F}\left(\lambda_{1:t}\right)∼N\left(m_{0}\left(\lambda_{1:t}\right),\Sigma_{0}\left(\lambda_{1:t},\lambda_{1:t}\right)\right).$$**S3.** Find the current optimal array placement $r^{*}$ corresponding to the ${\bar{F}}_{t}^{*}=\min_{s\leq t}\bar{F}\left(r_{1:t}\right)$.**S4.** Choose $r_{t+1}$ as the next sample point by finding the optimal value of optimization problem (Equation 11), (Equation 12) or (Equation 14) using conditional distribution (Equation 9).**S5.** Add $\left\{r_{t+1},\bar{F}\left(r_{t+1}\right)\right\}$ to the known sensor location sample, set t=t+1.**S6.** Repeat step 1, 2, 3, 4 and 5 until convergence.

## 4. Numerical Experiment

To demonstrate the algorithm’s performance, the microphone array placement design issue in different dimensions is provided. Convex optimization subproblems are solved using the quadprog function in Matlab, and Bayesian optimization pocket GpyOpt is employed in Python 3.7. All codes are performed on a laptop with Intel(R), Core(TM) i5 CPU, and 2.42 GHz.

In the following example, the desired response function is defined throughout an area that would be suitable for a hands-free or multimedia mobile phone application in the passband area:(15)$$G_{d}\left(s,\lambda ,f\right)=e^{-j2\pi f(\frac{∥s-r_{c}∥}{c}+\frac{L-1}{2}T)},$$
where $r_{c}=\sum_{i=1}^{M}r_{i}$ is the center position of all placement variables λ and c=340.9 m/s is the speed of sound in air. Equation (Equation 15) is from [6]. We put $G_{d}\left(\lambda ,s,f\right)=0$ in the stopband to remove the interference and background noise. The minimum distance between two distinct elements is ${\bar{\varepsilon }}_{d}=0.015^{2}$, ρ(s,f)=1, and $f_{s}=8$ kHz; maximum frequency is selected as 4 kHz. A performance limit, which is the logarithmic value of observations $\bar{F}\left(\lambda^{*}\right)$ under the optimal microphone array design $\lambda^{*}$, is provided to represent the differences among various microphone array configurations:$$\mathrm{PLIM}=:10\log F(\lambda^{*}).$$

### 4.1. The 2D Microphone Array Placement Design Problem

A two-dimensional microphone array configuration problem is considered firstly. Both the passband and the stopband are specified on the plane z=0, where the speaker is located. The microphones are located on plane z=1 (see Figure 2). Discussion and comparison of microphone arrays containing nine elements follow.

These are the specific region definitions:$$\Omega_{p}=\left\{\left(s,f\right)|∥\left(x,y\right)∥\leq 0.4\;m,z=0\;m,0.5\;\mathrm{kHz}\leq f\leq 1.5\;\mathrm{kHz}\right\},$$
and
$$\begin{array}{cc} \Omega_{s}=\; & \left\{(s,f)|∥(x,y)∥\leq 0.4\;m,z=0\;m,2.0\;\mathrm{kHz}\leq f\leq 4\;\mathrm{kHz}\right\}\\ & \cup \{(s,f)|1.8\;m\leq ∥(x,y)∥\leq 3.0\;m,z=0\;m,0.5\;\mathrm{kHz}\leq f\leq 1.5\;\mathrm{kHz}\}\\ & \cup \{(s,f)|1.8\;m\leq ∥(x,y)∥\leq 3.0\;m,z=0\;m,2.0\;\mathrm{kHz}\leq f\leq 4.0\;\mathrm{kHz}\}, \end{array}$$
and the placement feasible region is
Λ={λ||x|≤1.5m,|y|≤1.5m,z=1m}.

The discretization of $\Omega =\Omega_{p}\cup \Omega_{s}$ is applied: 60 points are generated for each frequency domain area and 0.2 m for each spatial domain region.

The performance and CPU time (measured in seconds) for the proposed Bayesian optimization method and the hybrid descent method applied to microphone arrays with nine elements are displayed in Table 1. In this table, the Bayesian optimization method consists mainly of a GP and different acquisition functions. It is evident that the proposed algorithm could considerably boost the speed of computing while achieving the same broadband beamformer performance as the hybrid descent approach [9]. The proposed algorithm reaches the optimal performance in only 1518 s, which is more than four times faster than the hybrid descent method. Moreover, an optimal set of microphone arrays can effectively improve the performance of the beamformer compared with linear placement. In Table 1, we also show the average stopband gain $G_{s}$ at f = 1400 Hz and filter length L=50. They illustrate that the noise in the stopband region is better suppressed with an optimal set of microphone array configurations compared with linear placement.

By applying the proposed algorithm and hybrid descent method, the optimal placement $r^{*}$ is shown in Figure 3 below. Because the optimization problem in (Equation 3) is non-convex, as can be seen from these figures, the microphone array configuration can vary significantly. Figure 4 shows the beamformer’s performance for the optimal microphone array configuration $r^{*}$ in the (x,y)-plane at 1400 Hz and in the (x,f)-plane at y=0 for a filter of finite length L=50 to illustrate the impact of the beamformer, where *G* denotes the beamformer’s output (Equation (Equation 1)) over the whole targeted region.

It is worth noting that a failure of one of the microphones in the optimal array configuration will not invalidate the beamformer, but it will degrade the performance. The stopband gain $G_{s}$ is −45.6333 dB in the optimal microphone array. If a microphone in the array fails randomly, stopband gain $G_{s}$ reduces to −31.7916 dB.

### 4.2. The 3D Microphone Array Placement Problem

In the next example, a 3D microphone array placement design problem is considered. The microphone elements are selected in a solid that is 0.5 m away from the desired cubic region for beamforming, as shown in Figure 5 below.

These are the specific region definitions:$$\Omega_{p}=\left\{\left(s,f\right)|∥\left(x,y\right)∥\leq 0.4\;m,1.5\;m\leq |z|\leq 2\;m,0.5\;\mathrm{kHz}\leq f\leq 1.5\;\mathrm{kHz}\right\},$$
and
$$\begin{array}{cc} \Omega_{s}=\; & \left\{(s,f)|∥(x,y)∥\leq 0.4\;m,1.5\;m\leq |z|\leq 2\;m,2.0\;\mathrm{kHz}\leq f\leq 4\;\mathrm{kHz}\right\}\\ & \cup \{(s,f)|∥(x,y)∥\geq 1.8\;m,|x|,|y|\leq 3.0\;m,1.5\;m\leq |z|\leq 2\;m,0.5\;\mathrm{kHz}\leq f\leq 1.5\;\mathrm{kHz}\}\\ & \cup \{(s,f)|1.8\;m\leq ∥(x,y)∥\leq 3.0\;m,1.5\;m\leq |z|\leq 2\;m,2.0\;\mathrm{kHz}\leq f\leq 4.0\;\mathrm{kHz}\}, \end{array}$$
and the placement feasible region is
Λ={λ||x|≤1.5m,|y|≤1.5m,1.5m≤|z|≤2m}.

The frequency and spatial domain areas are divided in a way that is similar to the 2D microphone array situation. Table 2 displays that the computational efficiency of the proposed algorithm can be increased by almost five times and that the performance of the beamformer is better than the hybrid descent method. Moreover, an optimal set of microphone arrays can effectively improve the performance of the beamformer compared with linear placement. The average stopband gain $G_{s}$ at 1400 Hz and z=1.6 m in Table 2 demonstrates that the noise at the stopband is suppressed well with the optimal array placement.

We primarily display in Figure 6 where the microphone arrays are located. It can be seen that the microphone placements are scattered evenly over the middle of the rectangle’s feasible domain, with a higher density on the side that is the furthest from the sound source. Figure 7 shows the beamformer’s performance for the ideal set of microphone arrays in the (x,y)-plane at 1400 Hz, z=1.6 m and in the (x,z)-plane at 1400 Hz, y=0 for a filter of finite length L=50.

The stopband gain $G_{S}$ that can be achieved with an optimal microphone array placement is −63.1721 dB. If an element in the microphone array fails, the stopband gain $G_{S}$ would reduce to −51.5854 dB without being completely ineffective.

In fact, we design 2D and 3D examples based on different scenarios. The 2D case is also a sub-example of the 3D one. In the 2D case, the microphone array is placed at the same level, but the microphones can be positioned at different levels in the 3D case. The 3D case is a bit more complex, and solving the optimization problem demands more computational time. In addition, better suppression of noise in the stopband is achieved in the 3D case.

## 5. Conclusions

In this paper, the Bayesian optimization method has been employed to solve the microphone array configuration design problem to enhance beamformer performance. Since the configuration design problem is non-convex and highly non-linear and the objective function is time-consuming to calculate, GP has been used as a surrogate function to approximate the objective function. The acquisition function guided the iterations toward the optimal set of microphone array placements in the sense of probability, and different acquisition functions were used for comparison. Numerical experiments have demonstrated that the proposed Bayesian optimization method finds similar or better microphone array configuration more efficiently. The proposed Bayesian optimization method is at least four times faster than the hybrid descent method to find the optimal placement from the numerical results. Therefore, the method is a competitive approach to design microphone placements when short time is required. As a future extension, it is interesting to consider alternative probabilistic agent models in Bayesian optimization to approximate the complicated objective function. Also, more advanced versions of acquisition functions could be considered. Optimal microphone arrays can be realized in many commercial products such as receivers in smart home systems and multi-function classrooms.

## Figures and Tables

**Figure 1 sensors-24-02434-f001:**
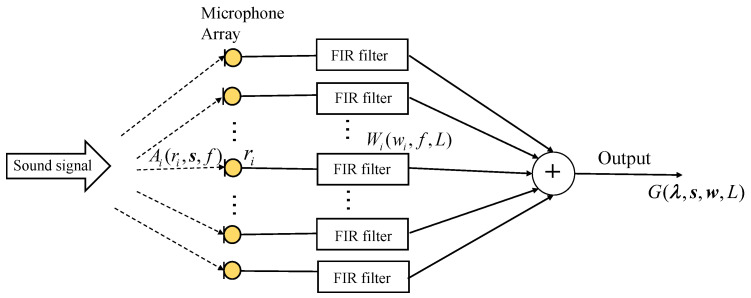
The structure of a microphone array with *N* microphones.

**Figure 2 sensors-24-02434-f002:**
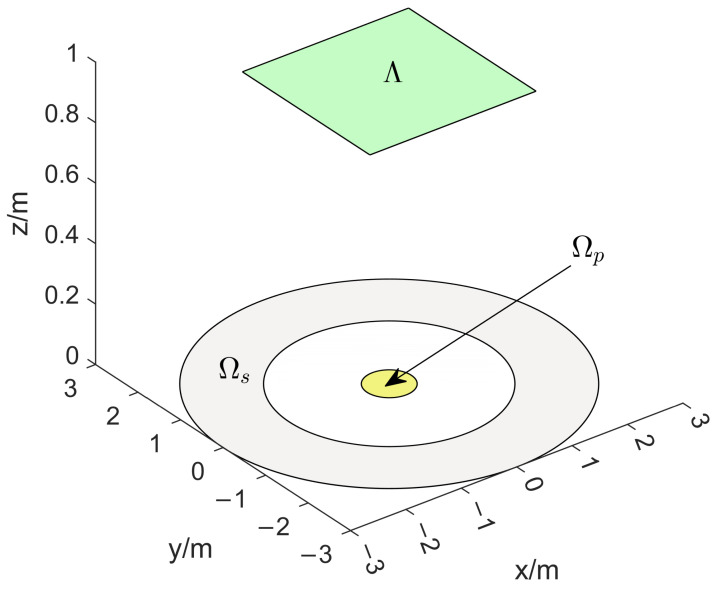
The setup of the 2D microphone array placement design problem.

**Figure 3 sensors-24-02434-f003:**
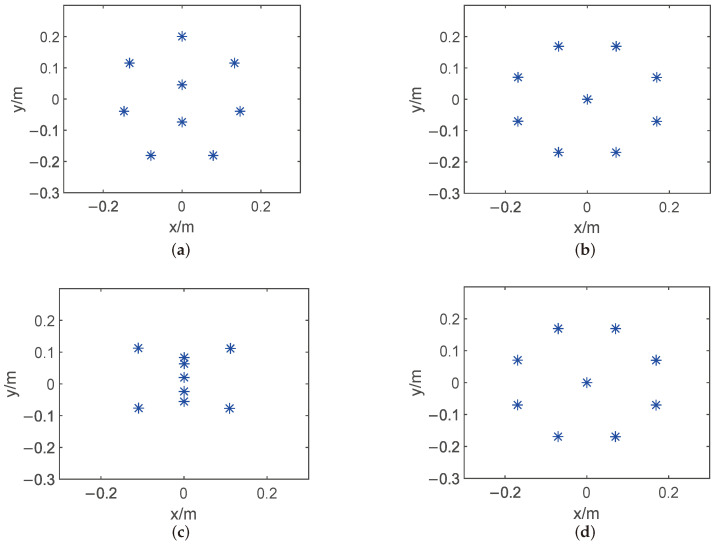
Optimal microphone array placement in the 2D plane with different methods, the blue stars represent the position of each microphone. (**a**) Bayesian optimization method based on GP regression and PI acquisition function. (**b**) Bayesian optimization method based on GP regression and EI acquisition function. (**c**) Bayesian optimization method based on GP regression and LCB acquisition function. (**d**) Hybrid descent method based on GA and gradient descent algorithm.

**Figure 4 sensors-24-02434-f004:**
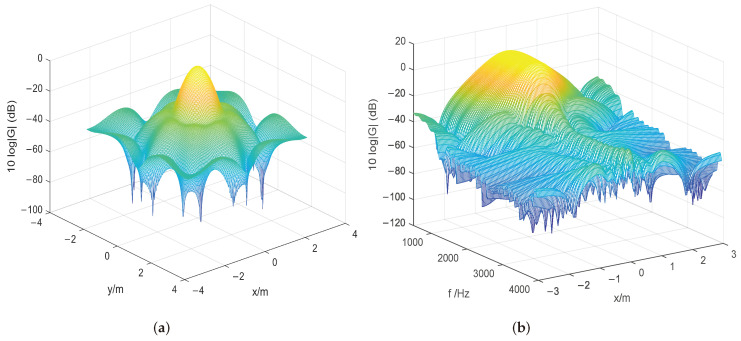
Performance under the optimal 2D configuration. (**a**) Beamformer output response in the (x, y)-plane at 1400 Hz with filter length L=50. (**b**) Beamformer output response in the (x,f)-plane at y=0 with filter length L=50.

**Figure 5 sensors-24-02434-f005:**
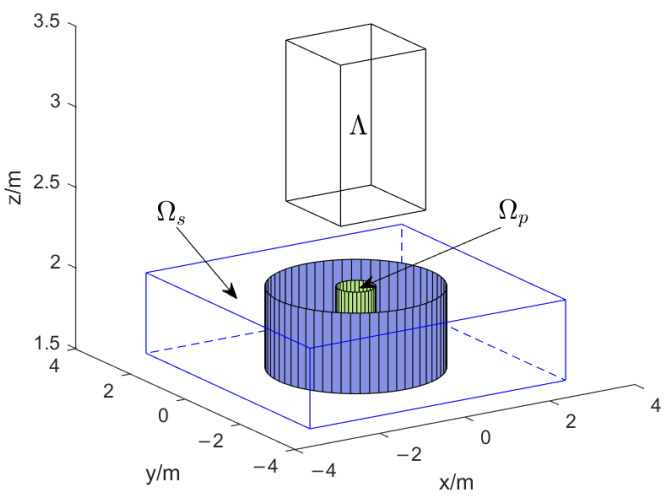
The setup of the 3D microphone array placement design problem.

**Figure 6 sensors-24-02434-f006:**
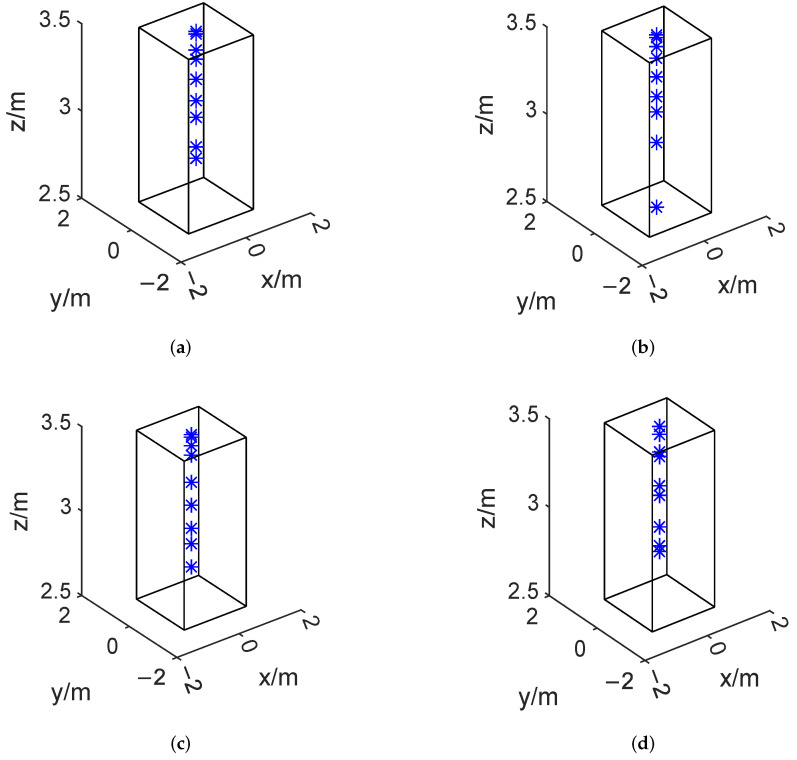
Optimal microphone array placement in the 3D plane with different methods, the blue stars represent the position of each microphone. (**a**) Bayesian optimization method based on GP regression and PI acquisition function. (**b**) Bayesian optimization method based on GP regression and EI acquisition function. (**c**) Bayesian optimization method based on GP regression and LCB acquisition function. (**d**) Hybrid descent method based on GA and gradient descent algorithm.

**Figure 7 sensors-24-02434-f007:**
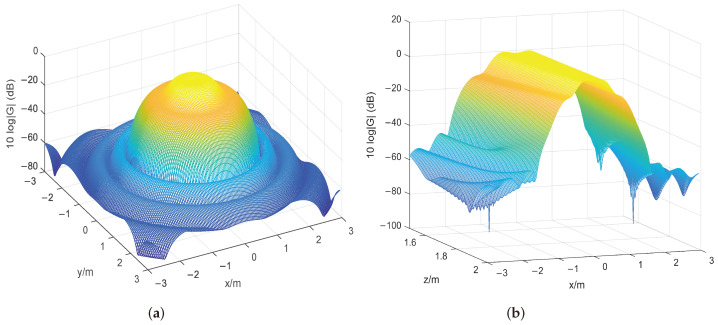
Performance under the optimal 3D configuration. (**a**) Beamformer output response in the (x,y)-plane at 1400 Hz, z=1.6 m, with filter length L=50. (**b**) Beamformer output response in the (x,z)-plane at 1400 Hz, y=0, with filter length L=50.

**Table 1 sensors-24-02434-t001:** Summary of beamformer performance with different array placement design.

	GP-PI	GP-EI	GP-LCB	GA-Gradient *	Linear
CPU time (s)	1255	1518	1375	6407	-
PLIM (dB)	−38.5996	−39.1635	−38.1337	−39.1635	−19.4577
$G_{s}$ (dB)	−43.5372	−45.6333	−42.0707	−45.6333	−25.0743

* GA-gradient stands for hybrid descent method.

**Table 2 sensors-24-02434-t002:** Summary of beamformer performance with different array placement designs.

	GP-PI	GP-EI	GP-LCB	GA-Gradient *	Linear
CPU time (s)	1718	1597	1535	8340	-
PLIM (dB)	−36.7066	−36.7088	−36.7082	−36.6520	−16.9972
$G_{s}$ (dB)	−61.2713	−62.7589	−63.1721	−61.7405	−27.4568

* GA-gradient stands for hybrid descent method.

## Data Availability

Data are contained within the article.

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
