# Peer review of "Optimal Microphone Array Placement Design Using the Bayesian Optimization Method"

_sensors, 2024, doi:10.3390/s24082434_

Round 1
Reviewer 1 Report
Comments and Suggestions for Authors
Submitted manuscript is devoted to the important subject – optimization of the placement of the array of microphones. Nowadays this technical problem appears in different ways and become a social and even quality of teaching/learning factors. However, the technical quality of the manuscript is low, it looks like a student laboratory exercise and neglect the style usually followed in MDPI Sensors. The Introduction must be finished by short and exact indication of the goal of the study. The diffused description of the research plan is not the scientific objective of the study (the last paragraph is not necessary and clear formulation of the objective of the study must be added.
The authors stay on the formal position about engineering optimization of the microphones positioning, however, in the majority of cases it depends on the set of available microphones (the same or not, the type of the unites of the arrays), the number of the sound sources (it is not the same for 2 persons interview or choir re-translation). In addition, discussion should include the analysis of critical failure probability for each configuration, including the operational failure of the switching for sound mixing console, power amplifier, columns etc.
There is no estimation for the development of the problem in the future – directions, problems, even analysis of the adaptation of the existing commercial products as soon as the objective is practical and engineering in its nature.
Tables have unrealistic indicators for significant numbers. All figures must show units for the axes.
Comments on the Quality of English LanguageQuite regular but understandable
Author Response
Many thanks for reviewer's comments, we benefitted greatly and have revised accordingly.

Reviewer 2 Report
Comments and Suggestions for Authors
Date: 3/18/2024
Journal: Sensors
Title: Optimal Microphone Array Placement Design Using Bayesian Optimization Method
Comments:
This work investigates the optimization problem of microphone array placement in beamforming. The authors propose a Bayesian optimization-based method to address this issue, which does not rely on local gradient and Hessian approximations but utilizes all information from prior evaluations. Through Gaussian process regression and acquisition functions, the method provides a probabilistic model for the objective function while integrating out uncertainty, and constructs a utility function from the model's posterior to determine the next evaluation point. Numerical experiments demonstrate that the method can find better microphone array placements more quickly, significantly improving computational efficiency. The paper also discusses the effectiveness and efficiency of the method in two numerical examples with different dimensions.
1. The introduction provides a good background on the importance of microphone array placement in beamforming. It would be beneficial to include a brief overview and a comparison of the algorithm based on prior knowledge.
2. Why is a microphone array with 9 elements used for simulation experiments? What is the basis for this choice?
3. In Figure 6, the 9 microphones are arranged in a vertical direction, which would not occur in reality.
Author Response

(The authors gave the same response as above.)

Reviewer 3 Report
Comments and Suggestions for Authors
1- Compassion table should be added and proposed results should be compared with some related works.
2- The abstract and Conclusion should be revised and supported by result.
3- The figures and equations which are not belong to authors should cited in the text.
4- Algorithm 1 (Bayesian optimization), should be provided as table or figure with table or figure number.
5- In 3D design when on dimension considered constant, 2D and 3D array placement should have same results, provide more explanations about comparison of these two cases.
6- The plagiarism check index (27%) is high and should be reduced.
7- Provide novelty of the proposed method clearly and state with numbers what improvement has been achieved compared to similar algorithms like.
8- Provide explanations about white noise problem, and improving the white noise gain (WNG) parameter in the proposed algorithm.
Comments on the Quality of English Language
Minor editing of English language required
Author Response

(The authors gave the same response as above.)

Round 2
Reviewer 1 Report
Comments and Suggestions for Authors
Authors must carefully revise the last part of the Introduction. It is not an industrial Lab. Report – there is no need to write the step-by-step plan of research but clearly formulated goal must be given.
All figure captions are too short and not informative. The format must be Figure caption:… (a) …(b). Figures 5,7,8 5 have 2 parts and no (a) and (b) description. Do not invent your own style – journal has clear indications for Authors.
Comments on the Quality of English LanguageAdditional proof-reading is necessary.
Reviewer 3 Report
Comments and Suggestions for Authors
The authors have addressed most of my concerns.
But the Abstract and conclusion are still not supported by the results and data.
Minor editing of English language required
